# Co-Extraction Technique Improves Functional Capacity and Health-Related Benefits of Olive Oils: A Mini Review

**DOI:** 10.3390/foods12081667

**Published:** 2023-04-17

**Authors:** Ítala M. G. Marx

**Affiliations:** 1Mountain Research Center (CIMO), Polytechnic Institute of Bragança, 5300-253 Bragança, Portugal; itala.marx@ipb.pt; 2Associated Laboratory for Sustainability and Technology in Mountain Regions (SusTEC), Polytechnic Institute of Bragança, 5300-253 Bragança, Portugal

**Keywords:** enriched olive oil, functional food, phenolic compounds, co-extraction technique

## Abstract

Olive oil, a fundamental component of the Mediterranean diet, is recognized as a functional food due to its health-promoting composition. The concentration of phenolic compounds in olive oil is influenced by various factors such as genetics, agro-climatic conditions, and technological processes. Therefore, to ensure an ideal intake of phenolics through the diet, it is recommended to produce functional enriched olive oil that contains a high concentration of bioactive compounds. The co-extraction technique is used to create innovative and differentiated products that promote the sensory and health-related composition of oils. To enrich olive oil, various natural sources of bioactive compounds can be used, including raw materials derived from the same olive tree such as olive leaves, as well as other compounds from plants and vegetables, such as herbs and spices (garlic, lemon, hot pepper, rosemary, thyme, and oregano). The development of functional enriched olive oils can contribute to the prevention of chronic diseases and improve consumers’ quality of life. This mini-review compiles and discusses relevant scientific information related to the development of enriched olive oil using the co-extraction technique and its positive effects on the health-related composition of oils.

## 1. Introduction

Virgin olive oil (VOO) plays a key role in the Mediterranean diet, being one of the most valued fats worldwide, with recognized health positive effects related to its bioactive composition [1].

Olive oil’s significance is attributed to its high levels of monounsaturated fatty acids and the presence of bioactive compounds such as α-tocopherol, phenolic compounds, chlorophyll, and carotenoids [2].

Tocopherols are naturally found in olive oil and are important indicators of its quality and stability. This is because they possess potent antioxidant properties, which prevent oxidative degradation during processing and storage. The levels of tocopherols in olive oil are influenced by various factors, such as cultivar, fruit ripeness, and extraction method. High-quality VOOs typically exhibit high α-tocopherol concentrations, which is the most biologically active form of vitamin E and possesses several health benefits, including prevention of certain types of cancer, heart disease, and other chronic ailments. Therefore, precise monitoring and control of the tocopherol content of VOO are essential to ensure its quality and nutritional value [3,4].

Moreover, chlorophyll and carotenoids are two classes of natural pigments that are found in plants and algae, including olives [5,6]. Chlorophyll is responsible for the green coloration of plants and is involved in photosynthesis. Carotenoids are responsible for the yellow, orange, and red colors found in many fruits and vegetables. In olive oil, these pigments play a critical role in determining its quality, stability, and nutritional value. Chlorophyll is a pro-oxidant and can accelerate the degradation of olive oil, while carotenoids possess potent antioxidant properties that protect the oil from oxidation and improve its shelf life. The concentration and composition of these pigments in olive oil can vary depending on several factors, such as the cultivar, fruit ripeness, and processing conditions. Extra virgin olive oils (EVOOs) are typically characterized by high levels of carotenoids and low levels of chlorophyll, indicating superior quality and freshness. Furthermore, the presence of these pigments in olive oil contributes to its sensory properties, such as flavor, aroma, and color. Therefore, monitoring and controlling the chlorophyll and carotenoid content of olive oil is crucial to ensure its quality, stability, and health benefits [4,5,6,7].

Phenols are among the most important nutraceutical compounds due to their nutritional and sensorial characteristics [8]. Phenolic compounds are natural antioxidants found in plants, including olives, and are responsible for many of the health benefits associated with olive oil consumption [9]. Commonly, oleuropein is the main phenolic compound in olives, which is easily extracted as part of the phenolic fraction of olive fruits, leaves, and seeds [10], presenting high biological properties, such as antioxidant and anti-inflammatory activities [11].

VOOs’ phenolic profiles and concentrations are influenced by the olive cultivar, ripeness index, harvesting methods, extraction methods, storage conditions, and environmental factors, such as temperature, rainfall, and soil quality, which can also affect the antioxidant levels in VOO [12,13].

The most abundant phenolic compounds in VOOs are hydroxytyrosol, tyrosol, oleuropein, and ligstroside derivatives, such as oleacein and oleocanthal. These compounds present a range of health-promoting properties, making them an important component of a healthy diet. Regular consumption of EVOO, which contains the highest levels of phenolic compounds, has been linked to reduced risk of chronic diseases such as cardiovascular disease, cancer, and neurodegenerative diseases [14,15,16].

Indeed, the consumption of phenolic compounds has been associated with the prevention of low-density lipoprotein lipoperoxidation, and daily intake of hydroxytyrosol and its derivatives is linked to an authorized health claim. The European Food Safety Authority (EFSA) has acknowledged that phenolic compounds in oils contribute to the protection of blood lipids against oxidative stress. However, this claim is only applicable to oils that contain a minimum of 5 mg of hydroxytyrosol and its derivatives per 20 g of oil [17].

Therefore, increasing oils’ natural antioxidant contents could contribute to improve chemical stability, while enhancing their nutritional and nutraceutical properties [1,18]. The development of functional foods may become a challenge, being necessary to identify new sources of bioactive compounds to increase the availability of healthy food products. In this sense, the use of natural sources of bioactive compounds to enrich oils in phenolic compounds seems to be a promising alternative to contribute to human health [18].

Functional enriched oils have gained significant attention in recent years due to their potential health-related benefits. Co-extraction techniques, such as those involving fruits, spices, and leaves, have been developed to enrich olive oil with additional bioactive compounds, potentially enhancing its nutraceutical properties [19,20]. However, the effectiveness of these co-extraction techniques in preserving the health-related composition of oil remains a topic of debate.

For olive oils producers, co-extracted oils could be an opportunity for innovation, differentiation, and creation of oils richer in bioactive composition. During the last years, some studies have shown oils co-extracted with the addition of olive leaves, bergamot, rosemary, thyme, basil, and oregano, to be responsible for increasing oils’ quality and positive sensory notes [1,19,21,22,23,24,25,26,27,28,29].

*Olea europaea* L. leaves are recognized as a significant by-product of olive farming. Numerous studies have demonstrated the high-added value of olive leaves due to their potent antioxidant properties, particularly their phenolic compounds, which exhibit robust preventive effects against oxidation [2,18,30].

Olive oil co-extraction with olive leaves is a technique used to enrich olive oil with additional bioactive compounds, such as phenolics. Olive leaves are recognized as a significant by-product of olive farming. Numerous studies have demonstrated the high-added value of olive leaves due to their potent antioxidant properties, particularly their phenolic compounds, which exhibit robust preventive effects against oxidation [2,18,30].

Several studies have investigated the effects of co-extraction of olive oil with olive leaves on the health-related composition of the resulting oil [2,9,27,28,31,32,33,34]. The results pointed that olive oils co-extracted with olive leaves significantly increased the phenolic content of the oil compared to traditional olive oil extraction methods.

In addition to its health benefits, co-extraction of oil with olive leaves can also have environmental benefits. Olive leaves are a by-product of the olive oil extraction process and are often discarded as waste. By co-extracting the leaves with the oil, this waste can be converted into a value-added product, reducing waste, and increasing the sustainability of the olive oil industry [35].

Co-extraction of olive oil with olive leaves is a promising technique for producing functional enriched olive oils with enhanced health benefits and minimal impact on the sensory properties of the oil. Further research is needed to fully understand the effects of co-extraction with olive leaves on the health-related composition of the oil and its potential applications in the olive oil industry.

In this sense, the aims of this mini review are: (i) to provide an overview of the influence of co-extraction technique on olive oil phenolic composition; (ii) to discuss the influence of natural sources of bioactive compounds associated parameters on the phenolic content of olive oils; and (iii) to describe the impact of the application of co-extraction technique during olive oil extraction on its health-related benefits linked to phenolic compounds.

## 2. Olive Oil Phenolic Composition

Phenolic compounds, the secondary metabolites, and the predominant groups of phytonutrients in plants, are highly valued for their bio-functional properties and defense mechanisms. These compounds are one of the most relevant chemical families found in olive oils [36]. Polyphenols are important olive oil minor constituents that presents health-related properties [37,38,39,40].

Phenolics also contribute to the organoleptic properties of VOO, they stimulate the taste receptors eliciting bitterness, pungency, and astringency perception [41]. Additionally, phenolic compounds protect VOO against oxidation because they scavenge free radicals, which cause the oxidative chain reaction and reduce the quality of olive oil [42].

Several studies deal not only with the nutritional effects of phenolics, but also with the agronomic factors that influence their presence in olive oil, the mechanism that contribute to oils’ shelf life and stability, and the importance of the processing conditions, as recently reviewed [15,43].

Oils’ phenolic compounds may be classified into five main categories [36,44]:(i)Secoiridoids;(ii)Phenolic alcohols;(iii)Phenolic acids;(iv)Flavones;(v)Lignans.

Due to the wide variety in the chemical makeup and phenolic characteristics of olive oils, the primary class of compounds responsible for these traits is secoiridoids [36]. Secoiridoids consist of a hydroxytyrosol or tyrosol molecule attached to elenolic acid or its derivatives, and are typically glycosylated. The most abundant glycoside found in olive fruit is oleuropein, but during fruit ripening and processing, the aglycone form is produced by the enzyme β-glucosidase.

Another secoiridoid compound of interest is ligstroside aglycon, and derivatives of both oleuropein aglycone and ligstroside aglycone have been detected as the main group of phenolic compounds in VOO. These compounds are synthesized in the olive fruit via the mevalonic acid pathway, which is influenced by ripening and interactions with microorganisms. They can also be produced via chemical and enzymatic reactions during the olive oil production process [45].

The secoiridoid biosynthetic pathway (Figure 1) is responsible for the production of phenolic compounds associated with health benefits, according to EFSA’s health claim. Lignans, formed by the condensation of aromatic aldehydes, have also been identified in oil, with pinoresinol and 1-Acetoxypinoresinol being the most abundant in VOO [45].

Flavonoids, which consist of two benzene rings connected by a linear three-carbon chain, are divided into several groups: flavones, flavonols, flavanones, and flavanols. Apigenin and luteolin, both flavones, are the most commonly found flavonoids in VOO [45].

Phenolic alcohols, characterized by the presence of a hydroxyl group attached to an aromatic hydrocarbon group, include hydroxytyrosol and tyrosol, which are the primary phenolic alcohols in olive oil [45]. While their concentrations are typically low in fresh samples, they increase with storage time due to the hydrolysis of secoiridoids (oleuropein and ligstroside aglycones). A health claim stating that the consumption of 5 mg of hydroxytyrosol and its derivatives per 20 g of olive oil could help prevent LDL oxidation has been made based on the observed health benefits of these compounds [17].

Phenolic acids found in VOO can be divided into two main groups: hydroxybenzoic acid derivatives (including p-hydroxybenzoic, protocatechuic, vanillic, syringic, and gallic acids) and hydroxycinnamic acid derivatives (including p-coumaric, ferulic, caffeic, and cinnamic acids) [45].

The formation of the main olive oils’ phenolic compounds could be explained by the biosynthetic pathway of secoiridoids (Figure 1), according to the information available in the literature. During the oil extraction process, oleuropein **4** and its biosynthetic precursor, ligstroside **3**, which are abundant in olives, suffer enzymatic hydrolysis mediated by enzymes present in the olive fruits.

The activity of β-glucosidase enzyme leads to the formation of ligstroside (p-HPEA-EDA) and oleuropein aglycons (3,4-DHPEA-EA) (**5** and **6**), which in turn can be converted to the respective mono-aldehydic forms (ligstroside and oleuropein aglycone mono-aldehydes) and to different diastereoisomers, due to the ring opening and ring closure of the hemiacetal group. The two dialdehydic forms, oleocanthal **11** (p-HPEA-EDA) and oleacein **12** (3,4-DHPEA-EDA), are formed by hydrolysis of the methyl ester followed by decarboxylation. The hydrolysis and decarboxylation reactions can occur either enzymatically or chemically due to the acidic aqueous conditions of the extraction. Recently, two methylesterase enzymes (elenolic acid methylesterase **E1** and **E2**) have been identified, being responsible for the conversion of the mono-aldehydic aglycones forms (from the β-glucosidase action on oleuropein and ligstroside) into oleacein and oleocanthal [47].

Over the years, numerous researchers have put forth biochemical pathways for the synthesis of secoiridoid derivatives, drawing from analyses of the phenolic profile of olive oil [48]. In the last decade, new forms of oleuropein and ligstroside aglycons named oleokoronal **9** and oleomissional **10** were presented after the analysis of olive oil [49]. In that same study, the researchers noted that when olive oil extracts interact with silica-based stationary phases, the aglycons can transform into other isomeric forms. This finding has led some recent literature to propose that the biosynthesis of oleocanthal and oleacein involves a two-step process: first, the hydrolysis of the glucosidic bond of ligstroside and oleuropein, catalyzed by β-glucosidase, leading to oleokoronal and oleomissional; and second, the removal of a methyl group by a methylesterase. The resulting carboxylic derivatives are unstable and can be readily decarboxylated upon heating, resulting in oleocanthal and oleacein (Figure 1).

The formation of phenolic alcohols, hydroxytyrosol **2** or tyrosol **1**, can occur through various pathways during the VOO extraction process and storage. One of these pathways involves the direct hydrolysis of oleuropein or ligstroside mediated by esterases, resulting in the formation of hydroxytyrosol/tyrosol and an oleoside methyl ester molecule. Another possibility is the esterase-catalyzed hydrolysis of the aglycon form, which produces hydroxytyrosol/tyrosol and elenolic acid [50]. Acid hydrolysis of oleuropein is also a potential pathway, resulting in the formation of hydroxytyrosol, elenolic acid, and glucose [51].

## 3. Health Benefits of VOO Phenolic Compounds

In the last decades, the literature has been proposing a correlation between the eating habits of people living around the Mediterranean area and low cardiovascular mortality. Among other dietary characteristics, the main particularity of the Mediterranean diet is that VOO is the primary lipids’ source. The health benefits attributed to the consumption of VOO are specifically related to EVOO, which is considered a key bioactive food because of its high nutritional quality [52]. As suggested by many researchers, most of the Mediterranean diet human health promotion effects can be attributed to VOO consumption [53,54].

During the past few years, the health properties of VOO have been intensely investigated and efficiently summarized in several reviews [37,38,39,55,56]. Recently, cumulative evidence have demonstrated that the minor components of VOO, mainly the phenolic compounds with antioxidant characteristics, may also contribute to the healthy features of VOO [14,43,57,58].

The human health-related benefits of minor VOO components are primarily due to phenols, which demonstrate a broad spectrum mainly of anti-inflammatory and antioxidant effects. Among the several chemical constituents of VOO, particular attention has been paid to phenolic compounds, largely due to their antioxidant effect, which has an approved health claim by the EFSA [59]. Most of oils’ health-related benefits involve antioxidant defense systems [60]. VOO consumption protects low density lipoproteins (LDLs) from being oxidized due to the binding capability of phenolic compounds to LDL particles [61].

Hydroxytyrosol is known for its ability to reduce oxidative stress when hydrogen peroxide is present. This phenolic compound can protect against oxidation by increasing the antioxidant activity and expression of both glutathione peroxidase and reductase [62]. Moreover, recent studies have shown that VOO lignans also exhibit antioxidant capacity in vitro [63]. The main mechanism of action of these phenolic compounds involves increasing the expression of antioxidant genes, scavenging free radicals, chelating metals, and improving mitochondrial function, among other functions [60,64,65].

Recently, a study conducted with oleacein and oleocanthal compounds, evaluated their cytoprotective potential. The researchers noticed that both compounds were found to promote cytoprotective pathways while suppressing oxidative stress [66]. In fact, VOOs phenolic compounds have been investigated as a potentially anti-aging due their antioxidant properties [65,67].

Additionally, several studies have been demonstrating that VOO consumption is responsible to promote anti-inflammatory effects [65,68,69]. Accumulating evidence over the years has suggested that regular consumption of VOO is associated with a reduced risk of developing chronic degenerative disorders such as cardiovascular diseases, type 2 diabetes, and cancer.

The health benefits of VOO have been attributed not only to its monounsaturated fats content but also to the presence of phenolic compounds that possess antioxidant, anti-inflammatory, and immunomodulatory properties. Studies have shown that the anti-inflammatory responses of phenolic compounds are accompanied by a downregulation of the expression of proinflammatory genes, low levels of proinflammatory proteins, as well as a lower total plasma/serum concentration of proinflammatory markers in both chronic and postprandial levels [14].

During the last few years, scientific papers have been reporting the correlation between VOO phenolic compound concentrations and the satisfactory plasma lipid response [70]. In this sense, a review was recently published that evaluated clinical trials of VOO and its effects on blood pressure [71], both experimental and human studies agree in showing anti-hypertensive effects of VOO. In addition, it was demonstrated that the consumption of a diet containing polyphenol-rich VOO can decrease blood pressure and improve endothelial function in young women with high-normal blood pressure or stage 1 essential hypertension [72]. The literature has shown that VOO polyphenols regulate the release of vasoconstrictors, vasodilators, and pro- and anti-inflammatory molecules through genomic mechanisms [73].

The neurodegenerative disorder known as dementia, is associated with a loss of these cognitive abilities. Oxidative stress and neuroinflammation are both associated with the pathophysiology of many degenerative diseases [74,75]. The literature has been described that VOO rich in phenolic compounds consumption including in the basis of Mediterranean diet, improves synaptic activity, short-term plasticity, memory, and neuropathology in a tauopathy model [76]. In addition, research has shown that hydroxytyrosol, oleuropein, and their derivatives, as well as oleocanthal, exhibit an anti-amyloid effect, providing protection against the cytotoxic effects of amyloid aggregates [77,78,79]. This may have a neuroprotective effect in diseases such as Alzheimer’s, which is characterized by amyloid deposition and autophagy impairment, contributing to a decrease in aggregated protein and a reduction in cognitive impairment models [54,76,79,80].

## 4. Technological Aspects That Affect VOO Phenolic Composition

International Olive Council (IOC) [81] defined VOOs, according to extraction, as “Oils which are obtained from the fruit of the olive tree (*Olea europaea* L.) solely by mechanical or other physical means under conditions, particularly thermal conditions, that do not lead to alterations in the oil, and which have not undergone any treatment other than washing, decantation, centrifugation, and filtration”. The legal limits regarding the chemical and sensory characteristics established by the EU regulation (Commission Regulation (EEC) Nº 2568/91) [82] and its following updates (EU) 2019/1604 [83], permit classifying olive oils into the categories: EVOO, VOO, and lampante olive oil.

VOO is produced solely through mechanical or physical processes [81], which involve a series of steps such as cleaning, crushing, malaxation, centrifugation, filtration, packaging, and storage. Each of these steps has an impact on the final product’s quality.

Malaxation involves two phases: fruit milling and paste malaxing. During industrial extraction of VOO, heat is commonly applied during paste malaxing to facilitate the coalescence of small oil droplets into larger ones, making them easier to separate using mechanical systems [84]. This process also breaks up oil/water emulsions formed during the crushing operation. The quality and quantity of oil extracted depend on various parameters, such as time, temperature, atmosphere, water, and co-adjuvants [85,86,87,88].

Centrifugation is a method used in VOO extraction to separate the oil fraction from the vegetable solid material and vegetation water [89]. There are several VOO extraction processes, including traditional pressing, the three-phase system, and the more recently developed two-phase centrifugal approach. The two-phase centrifugal method has become widely accepted for the production of high-quality VOOs. This method eliminates the use of water, and the resulting oil has a lower acidity and higher antioxidant content compared to other extraction techniques [90]. The two-phase continuous method prevails in Portugal, between 2018 and 2021 this method was responsible for 93% of national, and 89% of Trás-os-Montes region production [91].

Moreover, filtration is an important final step to remove suspended solids and moisture. The filtration process is recommended due to moisture reduction improving the oils’ quality. Higher polar phase content in unfiltered VOOs may increase alteration of oils’ quality [92].

During the past few years, consumers are increasingly interested in high-quality VOO with health and sensory properties linked to phenolic and volatile compounds. As a result, researchers are investigating novel and emerging technologies that can improve oil production. The extraction process is crucial in determining the quality and health properties of the oil. Therefore, optimizing malaxation conditions is of significant industrial importance to obtain high-quality oils.

## 5. Health Implications of Phenolic-Enriched Olive Oils

Despite the consumers’ increasing demand for products with high nutritional and health-related benefits, the olive oil industry has been actively pursuing product innovation to introduce novel olive oil-based products, such as functional phenolic-enriched olive oils. This area of research has gained momentum and is currently a topic of growing interest [18]. The enriched olive oil results from incorporating a natural source of bioactive compounds aiming to add inexistent compounds or group of compounds or enhance their contents, promoting specific nutritional or healthy characteristics [18,93].

In the next decade, approximately 70 million people around the word will be aged more than 65 years old [18]. The aged population increasing trend will imply a greater occurrence of chronic diseases of aging such as heart disease, cancer, osteoporosis, Alzheimer’s disease, among others, and will affect the health costs [94]. Indeed, the prevention of pathologies by diet is an important public health challenge to decrease diseases and mortality.

During the last years, the food industry has considered natural ingredients as a promising strategy to enhance food products health-related benefits. In this sense, the use of natural sources of bioactive compounds to enrich olive oils in phenolic compounds seems to be a promising alternative to contribute to human health [18].

Nowadays, innovation in the olive oils sector has included the increment of natural sources during the product extraction process, with the aim of improving their sensory and nutritional properties, as well as their stability [9,27,28,32,33,95]. The ingredients incremented may have various health benefits due to the presence of bioactive compounds with antioxidant, anti-inflammatory and/or antimicrobial properties [25]. These compounds play a protective character against oxidative stress, being also able to extend the oil shelf life [9]. *Olea europaea* L. by-products derived from the processing industry are secondary but important products, from which different bioactive compounds such as phenolic compounds, can be recovered and reused for numerous purposes following the circular economy policies [96]. In addition to olives by-products, aromatic herbs, fruits, and spices are the most conventional ingredients used, added either as an infusion, ethanolic extracts of essential oils, or by co-extraction [2,9,18,19,21,25,26,32,33,95,97].

Despite the various benefits mentioned in the literature, whether obtaining oils co-extracted with olive leaves, fruits, herbs, or spices, it should be kept in mind that only oils exclusively extracted from healthy olives can be commercialized as EVOO, according to the EU regulations [98]. Thus, these oils co-extracted in the presence of natural sources of bioactive compounds should not be commercialized as EVOO.

## 6. Co-Extraction Technique to Enrich Olive Oil

An innovative method to enrich olive oils in bioactive compounds, but still not widespread, is co-extraction, also known as co-milling or co-processing. This technique is based on the increment of natural ingredients, such as leaves, fruits, vegetables, or aromatic plants to the crushed olive paste before the malaxation step or during milling [19,20]; which may be implemented using ultrasound before the olive paste malaxation [25]. This technique allows for selecting both the type of olives, the ingredient incremented during the oils’ extraction with the greatest bioactive potential, as well as the extraction conditions. For olive oils producers, co-extracted oils could be an opportunity for innovation, differentiation, and creation of oils richer in bioactive composition.

During the last years, some studies have shown oils co-extracted with the addition of olive leaves, bergamot, rosemary, thyme, basil, and oregano, to be responsible for an increase in oils’ quality and positive sensory notes [1,19,21,22,23,24,25,26,27,28,29].

This method to enrich oils in bioactive compounds is an alternative technique to infusion, or to the addition of ethanolic extracts, to obtain enriched oils. The enrichment of olive oils by contact and essential oil incorporation are techniques implemented after the olive oil extraction, while enrichment by co-extraction implies the contact between olive mass and natural source of bioactive compounds during olive oil extraction. The co-extraction is an economically advantageous process for industry because a single process produces an optimal extractive yield in oil from different raw materials [22]. Additionally, a positive point of this method, when compared to the infusion one, is that the oil filtration step is not necessary [7].

In a recent study by Cecchi et al. [23] co-milling of olives and fresh chili peppers was compared to infusion of dried chili peppers in oil, using the same batch of olives for all oils. The researchers noted that oils prepared by co-milling were abundant in typical chili pepper esters, resulting in a pleasant hotness sensation and fresh pepper flavor. Moreover, the co-milling approach generally enabled the production of differentiated olive oil with improved sensory quality [23].

The proposed method to obtain an enriched olive oil is illustrated in Figure 2:

## 7. Bioactive Ingredients from Plant Materials to Enrich Olive Oil

Bioactive ingredients are responsible for providing some health benefits, which direct the food industry to focus its research on products of this nature [99]. In fact, phenolic compounds are one of the most studied bioactive compounds. Bioactive components of plant origin have excellent potential as functional food ingredients since they have health benefits. Thus, a diet that presents bioactive components is very important, and is a challenging area today, to improve health and consumers’ quality of life. In this sense, this section is meant to provide information on the addition of bioactive compounds from raw materials to improve olive oils bioactive composition through co-extraction process.

### 7.1. Olive Leaves to Enrich Olive Oil

During the past decades, oil consumption has experienced a constant increase, mainly attributed to its health-related and distinctive organoleptic characteristics. Therefore, waste and by-products resulting from olive production have also increased causing environmental problems and economic losses [100].

The activities involved in olive oil production generate significant amounts of by-products at various stages of the production process, including olive tree pruning, olive mill leaves, and olive pomace. These by-products are typically used for fertilizer purposes, which can have harmful environmental effects. However, due to their low cost and abundance, there is an opportunity for these by-products to be used for the development of high-value compounds, such as phenolics. This approach is known as valorization, which involves the recovery and transformation of by-products into useful materials, thereby reducing waste and generating new sources of income. In recent years, there has been an increasing interest in the valorization of olive oil by-products rich in bioactive compounds, such as phenolics, which have potential applications in the food, pharmaceutical, and cosmetic industries [101].

*Olea europaea* L. by-products, such as leaves, are considered as a cheap raw material that can be used as a useful source of phenolic compounds, exhibiting a strong protective effect against oils oxidation [102]. Olive leaves are not only accumulated during the agricultural and/or prunin activities but are hugely generated through the industrial activities of oil production, which account for 5% [103] up to 10% [104] of overall weight of olives harvested for processing.

The current literature is concentrated on the application of olive leaves as a powerful antioxidant source [105,106]. Phenolics classification is based on its structure as simple phenols, lignans, secoiridoids, flavonoids, flavonols, cinnamic acid derivatives, and phenolic acids [105,106]. The most abundant phenolic compound in olive leaves is oleuropein followed by hydroxytyrosol, and luteolin-7-O-glucosides [106].

Leaves of the olive tree contain various chemical compounds that can differ depending on factors such as cultivar, agro-climatic conditions, and selected extraction technology [107]. Despite being underutilized and typically used as animal feed [108], these by-products have considerable potential for high-value addition due to their abundance of bioactive compounds [103]. Olive leaves have been found to exhibit numerous biological activities, including antioxidant, antihypertensive, cardioprotective, and anti-inflammatory effects [109].

Additionally, olive leaves contain triterpenoids such as maslinic, ursolic, and oleanolic acids, which recent computational and in vitro studies have shown to possess anti-SARS-CoV-2 properties [110]. In vivo studies have also demonstrated the anti-inflammatory, analgesic, antipyretic, immunomodulatory, and antithrombotic activities of olive leaf extract, which are beneficial in controlling associated inflammatory cytokine storms and disseminated intravascular coagulation in COVID-19 patients. Given the biosafety, availability, and low cost of olive leaves, they are a promising candidate drug or supplement for controlling COVID-19 infection and warrant clinical investigation, as discussed in a recent study by Abdelgawad et al. [110].

Thus, olive leaf as a natural source of bioactive compounds is considered a potential alternative for enriching olive oils with health-related compounds. Several studies have reported that olive oils co-extracted with olive leaves can promote the oils’ phenolic, volatile, pigments content, and positive sensory attributes [2,9,27,28,31,32,33,34], as well as its anti-inflammatory, hypoglycemic, and antimicrobial properties [1]. The studies evaluated the impact of adding olive leaves (1–10%, *w*/*w*), dried or fresh, during laboratory, pilot, or industrial scale extraction, from different cultivars, including Cobrançosa, Oueslati, Neb Jmel, Moraiolo, Leccino, Buža and Arbequina [2,9,27,28,31,32,33,34]. However, it is important to emphasize that the results reported in the literature, vary according to several factors, such as the cultivar of the olive and the leaf, the amount of leaf used, as well as the oils’ extraction method applied (scale and conditions).

The impact of olive leaf addition on the chemical and sensory quality of extracted oils has been inconsistent across studies. The cultivar of olives and leaves, the proportion of leaves added, and the scale of extraction are important factors that strongly influence the resultant quality of oils.

Tarchoune et al. [9] conducted a laboratory-scale investigation to assess the impact of adding olive leaves (0% and 3%) on the primary antioxidants and antioxidant activity of Neb Jmel and Oueslati olive oils. The study used leaves and oils from Tunisian varieties harvested during the 2016/2017 crop season and analyzed for their concentrations of phenolics, tocopherols, and antioxidant power. The study found that Neb Jmel oil exhibited higher concentrations of chlorophyll (1.6-times), total phenolics (1.3-times), flavonoids (3-times), and oleuropein derivatives (1.5-times) than Oueslati oil. It also had a higher antioxidant activity (1.6-times). The addition of leaves resulted in a significant increase in total chlorophyll, α-tocopherol, and phenolics in both varieties, especially in Oueslati oil, due to the leaves’ higher abundance of bioactive constituents. The leaves had twice the pigment concentration of Neb Jmel leaves and three times the flavonoids and oleuropein derivatives. The study confirms that the Tunisian Neb Jmel and Oueslati varieties have oils with a chemical profile that meets the EVOO category and that their nutritional value increased with leaf addition [9].

Sanmartin et al. [33] enriched the nutraceutical qualities of EVOO by supplementing 3% (*w*/*w*) fresh olive leaves during oil extraction. The authors compared the chemical and sensory qualities of olive oils obtained from ripe olives pressed with either lemon, orange, or olive leaves to control EVOO. Olive oils extracted with olive leaves had a higher concentration of secoiridoids, 50% more than the control samples, as well as the greatest amount of chlorophylls. However, contradictory results were found when extracting oils from *cv.* Buža, with 1%, 2.5%, and 5% fresh olive leaves [33].

In their study, Novoselić et al. [32] investigated the impact of incorporating varying amounts of olive leaf (1%, 2.5%, and 5%, *w*/*w*) during the extraction of *cv.* Buža olive oil on its chemical composition and sensory properties. The authors found that the addition of olive leaves during the oil extraction process led to an increase in pigment concentration and intensified positive sensory attributes of the oil, such as fruitiness and green grass/leaf notes. However, the effect on oil phenolic composition varied depending on the amount of leaf added. When 1% of leaf was added, the majority of phenolic compounds were retained, while the addition of 5% leaf led to a decrease in the concentration of most phenols, particularly secoiridoids, by 45% compared to control samples. These findings suggest that the quantity of olive leaf present in the olive paste during oil extraction should be carefully considered, as higher amounts may have a negative impact on the phenolic profile of the resulting oil [32].

Marx et al. [27] conducted an industrial-scale study to extract oils from Arbequina olives with the addition of leaves from both Arbequina and Santulhana cultivars. The impact of adding leaves varied based on the cultivar used. Arbequina leaves increased the total phenolic content and total carotenoids content in the extracted oil, while Santulhana leaves resulted in a reduction in total phenolic content and chlorophyll content. Other studies have reported mixed results on the impact of adding leaves to olive oil extraction [9,33].

Marx et al. [27] also analyzed the oils’ phenolic profile and found that the addition of Arbequina leaves significantly increased the content of secoiridoid derivatives and total identified phenols. In contrast, adding Santulhana leaves decreased the total phenolic content and reduced the contents of oleuropein, oleacein, and oleocanthal derivatives in the extracted oil. These findings are consistent with previous studies that have examined the effect of adding leaves during oil extraction. For example, Ammar et al. [30] found an increase in hydroxytyrosol acetate and oleacein contents in oils extracted with the addition of leaves from the same cultivar.

The effects of olive leaf addition on the phenolic and pigment contents of olive oil were investigated by several studies. However, the results obtained by these studies were inconsistent. While Di Giovacchino et al. [34] reported a decrease in total phenolic content for Leccino and Castiglionese olive oils, Sevim and Tuncay [111] observed an increase in pilot-scale extraction. Similarly, the chlorophyll content was found to increase in industrial-scale extraction by Di Giovacchino et al. [34] and pilot-scale extraction by Sevim and Tuncay [111] (32–45% rise), Tarchoune et al. [9] (44–68% increase), and to a lesser extent (~1%) by Sanmartin et al. [33].

It is worth noting that the literature has mainly examined the impact of adding leaves to olives of the same cultivar on the composition of olive oils. Therefore, unlike the study conducted by Marx et al. [27], it is not possible to draw any conclusions about the potential effect of different leaf cultivars, which was found to be significant in their study.

These findings highlight the cultivar-specific nature of the results. The analyzed oils contained oleuropein and ligstroside derivatives, which serve as precursors to smaller molecules such as oleuropein, oleacein, and oleocanthal. However, the relative amounts of these derivatives varied among the oils. The activity of hydrolytic enzymes, such as β-glucosidase, can influence the phenolic fraction by releasing secoiridoid aglycones from their glucoside forms, while oxidative degradation catalyzed by PPO and POD can also play a role [27].

Marx et al. [27] observed a notable increase in phenolic content, particularly in the oleacein compound, in oils co-extracted with olive leaves at an industrial scale compared to control samples. However, opposite results were obtained in a study conducted at the laboratory scale (Abencor system), where the incorporation of olive leaves during oil extraction led to a significant reduction in total phenolic content. The authors attributed this trend to a reduction in the concentration of oleacein and oleocanthal, which followed the same relative reduction proportion as the total concentration of phenolic compounds [28]. The differences in the results may be due to the influence of leaves on enzymatic activities, which could be associated with factors such as the size of the added leaves, the method of incorporation, and the conditions and scale of the oil extraction process.

Finally, the contradictory trends observed in the literature, could be also attributed to leaves’ composition, the presence of microorganisms or the activity of enzymes present in leaves or secreted by the microorganisms present in the leaves [28,112].

In addition to the co-extraction process, recent studies have been reported the enrichment of olive oils in phenolic composition with olive leaf extracts [113,114]. However, although olive leaf extracts have demonstrated a potential for phenolic enrichment in oils, it is important to mention that this type of enrichment requires the process of extracting the phenolic compounds from the matrix, which requires organic solvents, and the application of techniques that require expensive equipment, such as lyophilization [101].

### 7.2. Herbs, Spices, and Fruits to Enrich Olive Oil

In recent years, there has been growing interest in using herbs and spices to enhance the bioactive compound content in oils, particularly olive oil. The sensory properties of these herbs and spices make them popular in the market, and they also have a positive impact on the nutritional value of enriched olive oils due to their high content of bioactive compounds. Co-extraction techniques have been used to study the effect of various herbs and spices on the bioactive compound content of oils, including garlic, lemon, oregano, hot pepper, and rosemary [26]; thyme and oregano [21]; *Thymus mastichina* L. [25] or grape and pomegranate seeds [22].

Baiano et al. [26] co-extracted olive oils with garlic, lemon, oregano, hot pepper, or rosemary. The results obtained demonstrated that the spices or fruits could increase the antioxidant capacity of oils, while their sensory proprieties are promoted [26].

*Thymus mastichina* L. was recently incorporated into the extraction process of *cv.* Galega oil, resulting in flavored oils that exhibited the characteristics VOO. However, due to regulatory constraints, these oils cannot be classified as VOO. Flavored oils obtained under optimized co-processing conditions (thyme concentrations > 3.5–4.0% and water contents ranging from 14 to 18%) displayed higher phenolic content and biological value than non-flavored VOO. Thyme flavor was strongly detected in the flavored oils, while the “wet wood” defect that is often present in VOO was not detected [25].

Compared to VOO and co-processed oil with thyme addition in the malaxator, the flavored oil produced by adding *T. mastichina* in the mill exhibited greater oxidative stability, even after six months of storage in the dark (16.6 vs. 10.3 h) [25].

Industrial by-products, such as grape and pomegranate seeds, are valuable sources of bioactive compounds that exhibit important health effects. These benefits arise mainly from their fatty acid composition. Grape seeds, for instance, contain about 7% fat, which is rich in linoleic acid, an essential polyunsaturated fatty acid. On the other hand, pomegranate seeds contain 7–16% fat, of which 65–80% is composed of conjugated fatty acids, with punicic acid (C18:3 n5), a conjugated linolenic acid (CLnA), being the most important one. In addition to these fatty acids, these seeds are also rich in tocopherols and tocotrienols, phytosterols, and polyphenols, which provide biological properties and prevent oxidation [22].

To investigate the potential of using grape and pomegranate seeds to enhance the nutritional value of oils, Marzocchi et al. [22] studied the effects of oils extracted from *cv.* Conicabra co-extracted with 25% (*w*/*w*) of grape and pomegranate seeds. The findings of their study showed that the by-products contain high levels of bioactive elements that could enrich the oil with healthy compounds such as punicic acid and γ-tocopherol. Therefore, the addition of grape and pomegranate seeds during the extraction process could lead to the production of oils with improved nutritional quality.

The Cornicabra cultivar was used to extract oils together with various fruits, spices, or leaves in the Abencor system to create novel olive oil-based products [20,115,116]. The extraction method applied was found to be more efficient in extracting phenolic compounds, resulting in a significantly lower hydrolysis of secoiridoids [19]. Additionally, the use of ultrasound technology inhibited the endogenous olive PPO enzyme, leading to significantly higher antioxidant activity compared to oils obtained by infusion technique [21].

Co-extracting olives with citrus fruits and/or zests resulted in the presence of new bioactive phenolics in the enriched olive oil, potentially contributing to the higher antioxidant activity observed in these oils compared to control oils [18,115].

However, the addition of fresh lemons during the olive milling process was found to negatively affect the quality parameters of the resulting oil, leading to a loss of phenolic compounds with antioxidant properties. This factor should be considered when producing enriched olive oil, and further research can be conducted to determine whether removing the lemon juice before processing the raw materials and adding only the lemon peel during the olive oil extraction process can lead to better quality oil [1].

The natural sources of bioactive components used to develop novel enriched olive oils are summarized in Table 1.

## 8. Concluding Remarks and Future Directions

Innovation in the oil sector is constantly growing. The development of functional enriched olive oil is opening new markets to the companies that have decided to make a leap of quality in the olive sector as an emerging strategy to improve the health-related effects of olive oil.

The results found in the current literature prove that food preferences impact health and, consequently, consumers want to choose healthy foods. Individualized nutrition permits in practice modifying enriched olive oils and other functional foods to individual needs seeking to improve their quality of life.

In this sense, the present work reviewed and discussed the literature reports about the effects of olive oil co-extraction with natural ingredients on the oils’ health-related composition. The results demonstrated that olive oils co-extracted with natural sources of bioactive compounds could be a promising method to enrich olive oils and promote health benefits for consumers, aiming to meet specific needs.

For this purpose, olive oil producers can bet on strategies to enrich olive oils using the co-extraction technique, aiming to obtain differentiated and innovative phenolic-enriched products.

Nonetheless, future studies need to be carried out to optimize the technological variables during the olive oil extraction process, such as malaxation time and temperature to ensure the highest concentration of natural compounds with bioactivity in the olive oil. Finally, after verifying the positive health effects of functional enriched olive oils, scientific research must be focused on determining the optimal concentration of these compounds with biological effects that should be incorporated to produce a functional enriched olive oil, with higher health-related composition.

## Figures and Tables

**Figure 1 foods-12-01667-f001:**
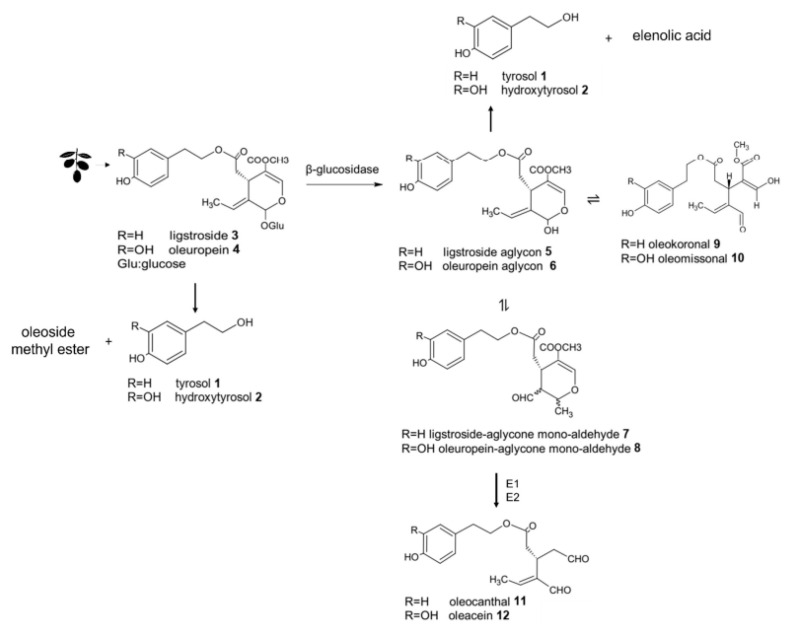
Formation pathway of some phenolics in VOO (biosynthetic pathway of secoiridoids), according to the information available in the literature [46].

**Figure 2 foods-12-01667-f002:**
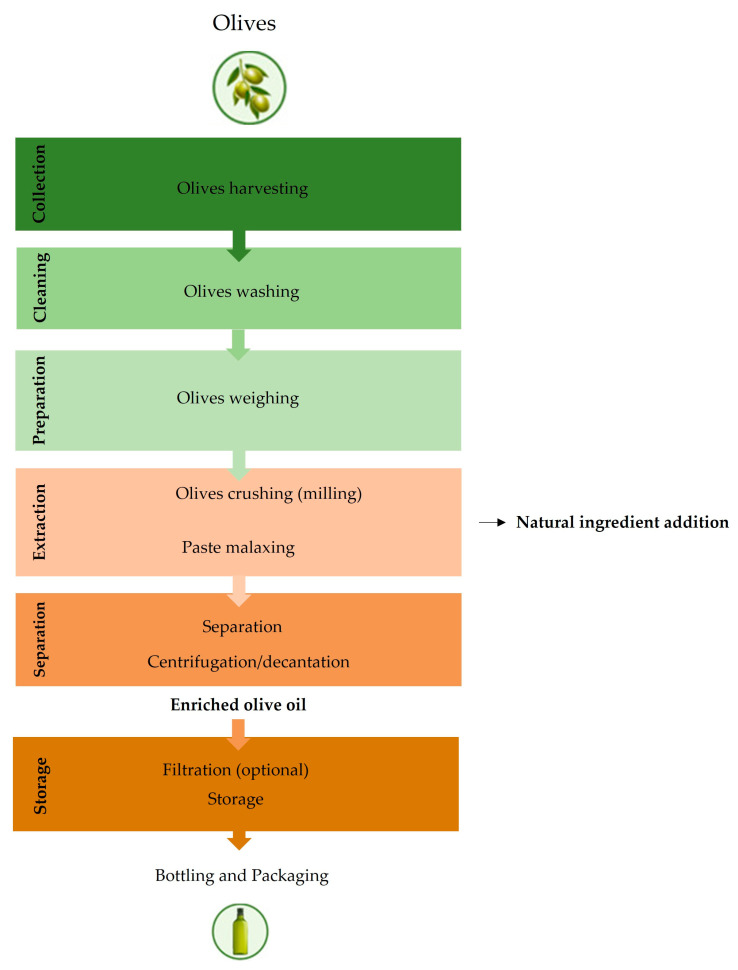
Scheme of enriched olive oil production process, according to merging information found in the literature [27,28,29].

**Table 1 foods-12-01667-t001:** Literature summary of the studies reporting olive oil co-extraction with natural bioactive ingredients and its effects on oils health-related composition.

Natural Bioactive Ingredient	Amount Added (%)	Olive Cultivar	Main Findings	Reference
**Olive leaves**	3% (*w*/*w*) of fresh leaves	Oils co-extracted from: - *cv.* Neb Jmel olives and leaves - *cv*. Oueslati olives and leaves	↑ resistance to oxidation ↑ total chlorophyll ↑ total carotenoids ↑ α-tocopherol ↑ oleuropein derivatives contents ↑ total phenolic contents	[9]
	3% (*w*/*w*) of fresh leaves	Oils co-extracted from: *-cv.* Moraiolo olives and leaves - *cv.* Leccino olives and leaves	↑ oleuropein aglycone	[33]
	1, 2.5, 5, and 10% (*w*/*w*) of fresh leaves	Oils co-extracted from: -*cv.* Cobrançosa olives and leaves	↑ α-tocopherol ↑ total chlorophyll ↑ total carotenoids	[2]
	1, 2.5, and 5% (*w*/*w*) of fresh leaves	Oils co-extracted from: *- cv.* Buža olives and leaves	↑ total chlorophyll ↑ total carotenoids ↓ phenolic compounds	[32]
	1–5% (*w*/*w*) of fresh leaves	Oils co-extracted from: *- cv.* Dritta olives and leaves *- cv.* Leccino olives and leaves *- cv.* Castiglionese olives and leaves	↑total chlorophyll ↑ total carotenoids ↑ total phenolic content	[34]
	3% (*w*/*w*) of fresh leaves	Oils co-extracted from: *- cvs*. Chemlali, Chétoui, Zalmati, and crossbreeding Chemlali by Zalmati - with *cvs.* Chemlali, Chétoui, Zalmati, and crossbreeding Chemlali by Zalmati leaves	↑ total tocopherols ↑ polyunsaturated fatty acids	[117]
	1% (*w*/*w*) of fresh leaves	Oils co-extracted from: *- cv.* Arbequina olives with *cv.* Arbequina and Santulhana leaves	↑ phenolic content ↑ total phenolic content ↑ oxidative stability ↑ total chlorophyll ↑ total carotenoids	[27]
**Tomato**	0.45–2.5% (*w*/*w*)	Oils co-extracted from *cv.* Correggiolo and blend of *cvs.* Moraiolo, Leccino and Correggiolo with tomato seed or skin	↑ phenolic content ↑ carotenoids ↑ lycopene ↑ β-carotene ↑ tocopherols	[24]
**Thyme and oregano**	1% (*w*/*w*)	Oils co-extracted from *cv.* Coratina with thyme and oregano	↑ antioxidant capacity ↑ total phenolic content	[21]
**Garlic** **Lemon** **Oregano** **Hot pepper** **Rosemary**	2–20% (*w*/*w*)	Oils co-extracted from *cv*. Peranzana with - garlic - lemon - oregano - hot pepper - rosemary	↓ total phenolic content ↑ antioxidant activity	[26]
***Thymus mastichina* L.**	0.4−4.6% (*w*/*w*)	Oils co-extracted from *cv.* Galega Vulgar with *Thymus mastichia* L.	↑ oxidative stability	[25]
**Grape and pomegranate seeds**	25% (*w*/*w*)	Oils co-extracted from *cv.* Coratina with - grape seeds - pomegranate seeds	↑ punicic acid ↑ γ-tocopherol ↓ total phenolic content	[22]

↑ Increasing trend; ↓ decreasing trend.

## Data Availability

Not applicable.

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
