# Peer review of "Co-Extraction Technique Improves Functional Capacity and Health-Related Benefits of Olive Oils: A Mini Review"

_foods, 2023, doi:10.3390/foods12081667_

Round 1

Reviewer 1 Report

The current manuscript entitled "Functional enriched olive oils: Effects of co-extraction technique on their health-related composition " is about the interesting topic of improving bioactives in the food product. The manuscript is well-written. However, a graphical abstract is needed to schematically illustrate the effects of co-extraction from different sources on olive oils. This manuscript should consider the following minor corrections.

Author Response

Dear reviewer, responses to all suggestions are attached.

Reviewer 2 Report

Comments to authors

Foods-2288523

Title: Functional enriched olive oils: Effects of co-extraction technique on their health-related composition

The manuscript described the effects of the co-extraction of olive oil and its health-related effects in detail. However, a table that summarizes the benefits of olive oil and a comparison of other techniques with the reviewed technique should be given, and the demand for the usage of enriched olive oil in the future and the development direction of the technology should be pointed out. This manuscript may be helpful to some researchers. However, there are some inappropriate statements that should be revised.

1.     The introduction is not well written. Authors should clarify the relationship between the functionality of olive oil, the effectiveness of the extraction technique, and its applications, especially the introduction of application potential is confused. So, the importance of the current study should be stated. The authors have only discussed the phenolic compounds and their benefits, while the review is on the functionality and extraction of olive oil. The introduction should be improved by discussing the components mentioned in the title and study.

2.     The sequence of paragraphs is not consistent

3.     Line 33-36 Please check this sentence.

4.     Lines 42-55, lines must be justified.

5.     Line 42, what's the meaning of "protect against oxidative stress"?

6.     Line 47-49, re-write the statement.

7.     Line 50, is grammatically incorrect. Please revise it with the correct language.

8.     Lines 211-215, are confusing and vague.

9.     The repetition of information discussed above should be avoided. Carefully read the manuscript and do not repeat the information.

Author Response

(The authors gave the same response as above.)

Reviewer 3 Report

The present minireview shows the importance of the production of functional olive oil enriched with polyphenols from several vegetable source, olive leaves included.

In my opinion, the paper must be deeply improved.

First of all, in the introduction, this paper didn’t consider all the aspects which might influence the qualitative quantitative presence of antioxidant in a virgin olive oil. In fact, only few words were dedicated to some and not all the variables WHICH STRONGLY INFLUENCE THE DEFINITIVE CONTENT OF SECORIDOIDS DERIVATIVES and tocopherols IN THE VIRGIN OLIVE OIL. The author should know that the entire extraction process  could compromise the final concentration of such bioactive compounds like the proper extraction method,  the filtration process as well as the bottling phase, the types of packaging used as well as the shelf life parameters adopted.

I would like to recognize the existence  of several studies on these topics, in particular on the packaging effect and the shelf life conditions  like the light exposure or temperature.

In this regard, I would like to underline that the author made the same error when described the different types of coextraction processes. Even in that case, in fact, the author didn’t report the evolution of these enriched oils after production which, I think is one of the most important aspect to understand, since its potential importance from a health point of view.

Furthermore, I would like to remember the author that oleuropein and ligstroside are the main secoiridoids present in the olive fruit and not in the EVOO, which in fact, contains, its derivatives!!!

Finally, since this paper is a minireview, I would have appreciated more figures and tables to have an immediate impact about the topic.  Only a table and a very simple scheme was reported in the entire manuscript.

No figures on the chemical structures of the substances indagated

No tables on the sensory and health properties of the compounds considered

Moreover, I found that the manuscript is not easy to read because of a certain confusion which the author made in the first part of the minireview starting to describe the eventual use of olive leaf, that is, in reality one of the main ingredients considered  successively in a proper paragraph.

Author Response

(The authors gave the same response as above.)

Reviewer 4 Report

This review analyse, at laboratory- or industrial-scale studies, the possibility to increase the functional properties of olive oils by co-extraction with natural bioactive ingredients in order to improve oils health-related composition. Even waste material such as olive leaves was taken into consideration giving a sustainability aspect to the process.

Overall, the review is well-written and major significant literature is analysed. The topic is of interest and falls within the scope of the journal. Conclusions are pertinent.

 Minor:

-  Lines 90, 94, 175, 230: revise literature cited and update;

-        -  Line 330: put abbreviations in full.

Author Response

(The authors gave the same response as above.)

Reviewer 5 Report

I believe the review is quite similar to a recent review published by Lamas, S., Rodrigues, N., Peres, A. M., & Pereira, J. A. (2022). Flavoured and fortified olive oils-Pros and cons. Trends in Food Science & Technology.

The author says that the review is a mini-review and it seems that it is not strong and systematic. i 

Author Response

(The authors gave the same response as above.)
